# LETTERS

## OPEN
# Airway-resident T cells from unexposed individuals cross-recognize SARS-CoV-2

Mariana O. Diniz[1,5], Elena Mitsi [2,5], Leo Swadling [1], Jamie Rylance [2], Marina Johnson[3], David Goldblatt [3], Daniela Ferreira [2,4,6] and Mala K. Maini [1,6]

**T cells can contribute to clearance of respiratory viruses that cause acute-resolving infections such as SARS-CoV-2, helping to provide long-lived protection against disease. Recent studies have suggested an additional role for T cells in resisting overt infection: pre-existing cross-reactive responses were preferentially enriched in healthcare workers who had abortive infections[1], and in household contacts protected from infection[2]. We hypothesize that such early viral control would require pre-existing cross-reactive memory T cells already resident at the site of infection; such airway-resident responses have been shown to be critical for mediating protection after intranasal vaccination in a murine model of SARS-CoV[3]. Bronchoalveolar lavage samples from the lower respiratory tract of healthy donors obtained before the COVID-19 pandemic revealed airway-resident, SARS-CoV-2-cross-reactive T cells, which correlated with the strength of human seasonal coronavirus immunity. We therefore demonstrate the potential to harness functional airway-resident SARS-CoV-2-reactive T cells in next-generation mucosal vaccines.**

To examine for severe acute respiratory syndrome coronavirus 2 (SARS-CoV-2) cross-reactive T cells in the lower respiratory tract airways, we used cryopreserved BAL samples taken during bronchoscopy of ten healthy donors in 2016–2018, before the onset of the pandemic. Cryopreserved peripheral blood mononuclear cells (PBMCs) from paired blood samples were also analyzed. Donors had been challenged with *Streptococcus pneumoniae* and influenza vaccine 6–19 weeks before bronchoscopy. Oropharyngeal swabs or nasosorption samples used for PCR screening of multiple respiratory viruses were negative in all subjects, apart from rhinovirus in one (patient cohort details in Methods and Supplementary Table 1). Mononuclear cells from BAL were stimulated with pools of SARS-CoV-2 peptides of specificities previously associated with protective pre-existing T cells[1,4]: three pools of overlapping peptides spanning the core replication transcription complex (RTC) non-structural proteins (NSPs) NSP7, NSP12 and NSP13 from *ORF1ab* and a pool of predicted epitopes from the structural protein spike. After overnight peptide stimulation, antigen-specific T cells with antiviral functional potential were enumerated by intracellular cytokine staining for interferon (IFN)-γ and/or tumor necrosis factor (TNF) (CD8+ and CD4+) and CD40L (CD4+), after subtracting the background effector function seen in the donor-matched unstimulated control well (<0.6% background cytokine production in all cases, gating strategy, example FACS plots and all background values in Extended Data Fig. 1a,b).

SARS-CoV-2 peptide-reactive CD4+ and CD8+ T cells were detected in six of ten of the pre-pandemic BAL samples (Fig. 1a,b and Extended Data Fig. 1c). Responses were detectable against each of the RTC regions, including the highly conserved RNA polymerase (NSP12)[5,6] that we found to be most strongly associated with abortive infection, as well as spike. SARS-CoV-2 cross-reactive CD4+ T cells were present in BAL at higher frequencies than CD8+ T cells (Fig. 1c), as noted in previous studies of the periphery[1,7–10]. However, CD4+ and CD8+ T cell responses were correlated in frequency within the same donor BALs (Extended Data Fig. 1d), indicating a coordinated response. Lung CD4+ and CD8+ T cells responded to SARS-CoV-2 peptides with induction of TNF and IFN-γ, with TNF being more abundantly produced than IFN-γ (Fig. 1d). Co-staining showed that some responding cells were multifunctional, with most IFN-γ-producing CD4+ and CD8+ T cells also producing TNF and a proportion of IFN-γ- and TNF-producing CD4+ T cells co-expressing CD40L (Extended Data Fig. 1e–h). The proportion of CD4+ T cells producing TNF was particularly high in some donors, but they were proportional to IFN-γ+CD4+ T cell and multifunctional (TNF/IFN-γ or TNF/CD40L CD4+ T cell) responses (Extended Data Fig. 1h and Fig. 1e), supporting antigen-specific triggering, perhaps amplified by a bystander TNF response[11].

BAL samples from the lower respiratory tract would be expected to contain some tissue-resident memory T cells ($T_{RM}$) that are specialized for long-lived sentinel pathogen defence[12]; to examine this we compared the residency profile (CD69/CD103 expression) of global T cells in BAL with paired PBMCs from the same donors. The *t*-distributed stochastic neighbor embedding (*t*-SNE) analysis confirmed differential segregation of cells within PBMCs and BAL expressing residency markers, particularly for CD69+CD103+ co-expression in the global CD8+ pool and single-positive CD69+CD103− in the CD4+ compartment (Fig. 2a,b). Among PBMCs, only 7.6% of global CD4+ T cells expressed CD69 compared with 66.7% in BAL (Fig. 2c,d). As previously reported, only a small population of airway CD69+CD4+ T cells co-expressed CD103 (ref. [13]). Circulating CD8 +T cells were predominantly CD69−CD103−, whereas CD8+ T cells with the residency phenotypes CD69+CD103− and CD69+CD103+ dominated in BAL samples (Fig. 2c,d).

In four BAL donors we detected virus-specific T cells of sufficient magnitude to assess their residency phenotype; SARS-CoV-2 RTC and spike-specific CD4+ T cells tended to have even higher frequencies of CD69 expression than global BAL T cells, especially among the IFN-γ+ population, where ~90% expressed CD69 (Fig. 2e and Extended Data Fig. 2a). Similarly, a high proportion of

[1]Division of Infection and Immunity and Institute of Immunity and Transplantation, UCL, London, UK. [2]Department of Clinical Science, Liverpool School of Tropical Medicine, Liverpool, UK. [3]Institute of Child Health, London, UK. [4]Present address: Oxford Vaccine Group, Department of Paediatrics, University of Oxford, Oxford, UK. [5]These authors contributed equally: Mariana O. Diniz, Elena Mitsi. [6]These authors jointly supervised this work: Daniela Ferreira, Mala K. Maini. ✉e-mail: Daniela.Ferreira@lstmed.ac.uk; m.maini@ucl.ac.uk

SARS-CoV-2-specific CD8[+] T cells had a residency signature (mean 70.6% CD69[+]CD103[+], 97% CD69[+]; Fig. 2f). In this limited sample size, the expression of residency markers was comparable across all antigen specificities examined (Extended Data Fig. 2a,b). Overall, the frequencies of SARS-CoV-2-specific T cells were significantly enriched within the CD4[+] and CD8[+] T_RM cell subsets compared with the non-T_RM cell pools in the airways (Fig. 2g,h and Extended Data Fig. 2a,b).

Tissue-residency provides a mechanism to retain an enriched frequency of virus-specific memory T cells at the site of antigen encounter. Having identified SARS-CoV-2 RTC and spike-cross-reactive T cells with a T_RM cell phenotype in the lower airways, we postulated that they would be selectively enriched at this site compared with the circulation. To examine whether pre-existing SARS-CoV-2-cross-reactive T cells were selectively enriched at the site of infection, we compared their frequencies after overnight stimulation of paired blood and BAL samples from the same donors. The percentage of CD4[+] and CD8[+] T cells producing TNF or IFN-γ (Fig. 3a,b) in response to the four SARS-CoV-2 peptide pools tested was substantially higher within BAL than PBMCs. CD4[+] and CD8[+] T cells tended to have broader specificity (responding to more peptide pools) in BAL compared with PBMCs (Fig. 3c,d). In addition, a higher proportion of donors had a detectable response to each peptide pool in their airway than in their blood sample (Fig. 3e,f). Overnight stimulation did not result in high expression of CD69 among peptide-responsive populations in the periphery (Extended Data Fig. 3a,b). This contrasted with the high CD69 expression of peptide-specific T cells described above in the airway, reinforcing the latter being a feature of T_RM cells rather than simply a result of peptide activation.

Next we examined whether circulating SARS-COV-2-reactive T cells identified after either overnight stimulation or in vitro expansion correlated with BAL responses. RTC and spike-cross-reactive T cells were detectable in more PBMC samples after short-term (10 d) in vitro expansion than overnight, consistent with these low-magnitude responses being below the threshold of detection rather than completely absent from the circulation (Extended Data Fig. 3c,d). Generally, overnight or in vitro expanded SARS-CoV-2-reactive T cell frequencies did not correlate significantly with those in matched BAL (Extended Data Fig. 3e–h), which may reflect the small sample size, but suggests that the magnitude of circulating T cells is not fully representative of the pool compartmentalized within the lung. Notably, the only responses to significantly correlate between blood and BAL were overnight IFN-γ[+]CD4[+] T cells specific for RNA polymerase (NSP12; Extended Data Fig. 3f), the region most highly conserved across human coronaviruses, supporting our previous association of this specificity in the circulation with protection from overt infection in healthcare workers[1]. Moreover, overnight peptide stimulation of PBMCs revealed significantly increased CD4[+] and a trend to higher frequencies of CD8[+] T cell responses in individuals with a detectable BAL response to SARS-CoV-2 (Fig. 3g,h).

Finally, we investigated the postulate that seasonal human coronaviruses (HCoVs) represent one probable stimulus for pre-existing SARS-CoV-2-reactive airway T cells. As insufficient BAL remained, we used residual paired PBMCs and sera to test for pre-existing T cell and humoral immunity to seasonal HCoVs. We have shown that 15-mer peptides spanning the nonstructural RTC region are highly conserved between HCoV and SARS-CoV-2 (ref. [1]). To assess the crossreactivity of the pre-existing SARS-CoV-2-reactive T cells to HCoVs, we therefore focused on spike, constructing a mapped epitope pool of peptides from the HCoV spike (OC43, 229E and NL63; BEI Resources, peptide arrays, 17-mers), equivalent to the pool we had used to assess SARS-CoV-2 spike-reactive T cells. Both overnight and in vitro expanded CD4[+] and CD8[+] T cell responses

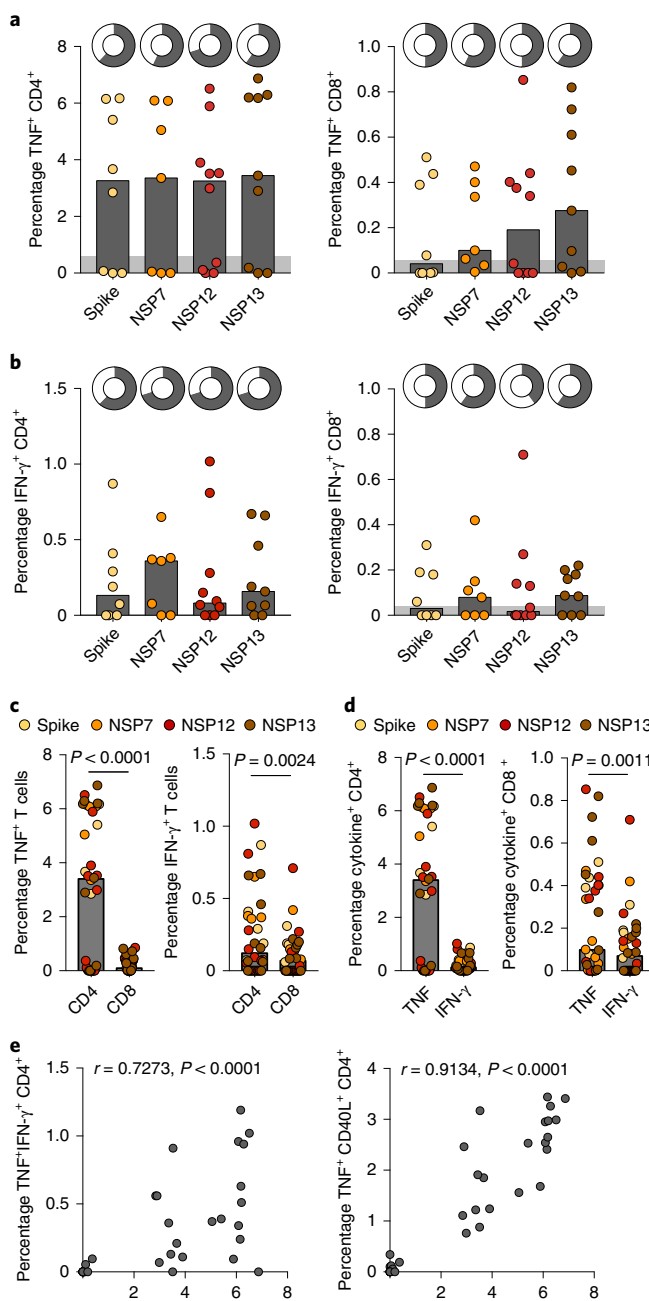

**Fig. 1 | Crossreactive SARS-CoV-2-specific T cells are present in pre-pandemic BAL samples. a,b**, Frequency of TNF- (**a**) and IFN-γ-producing (**b**) CD4[+] and CD8[+] T cells in BAL for each peptide pool. Doughnut plots at the top show the percentage of detectable responses. **c**, Percentage of TNF- or IFN-γ-producing CD4[+] or CD8[+] T cells in BAL. **d**, Frequencies of Sars-CoV-2-specific CD4[+] and CD8[+] T cells producing TNF or IFN-γ. **e**, Correlation of TNF[+] CD4 T cells versus TNF[+]IFN-γ[+] (left) and CD40L[+]TNF[+] (right) CD4 T cells in BAL (n = 10 biologically independent samples examined over one independent experiment). Bars at median (**a–d**), with gray area representing mean + 2 s.d. of DMSO control; Kruskal–Wallis one-way ANOVA test and Dunn's multiple comparison (**a** and **b**); Wilcoxon's paired test (**c** and **d**); Spearman's correlation (**e**).

to the seasonal HCoV spike tended to be higher frequency (significantly so for IFN-γ[+]CD8[+] T cells) in those pre-pandemic donors with SARS-CoV-2-reactive T cells in their BAL (Fig. 4a,b).

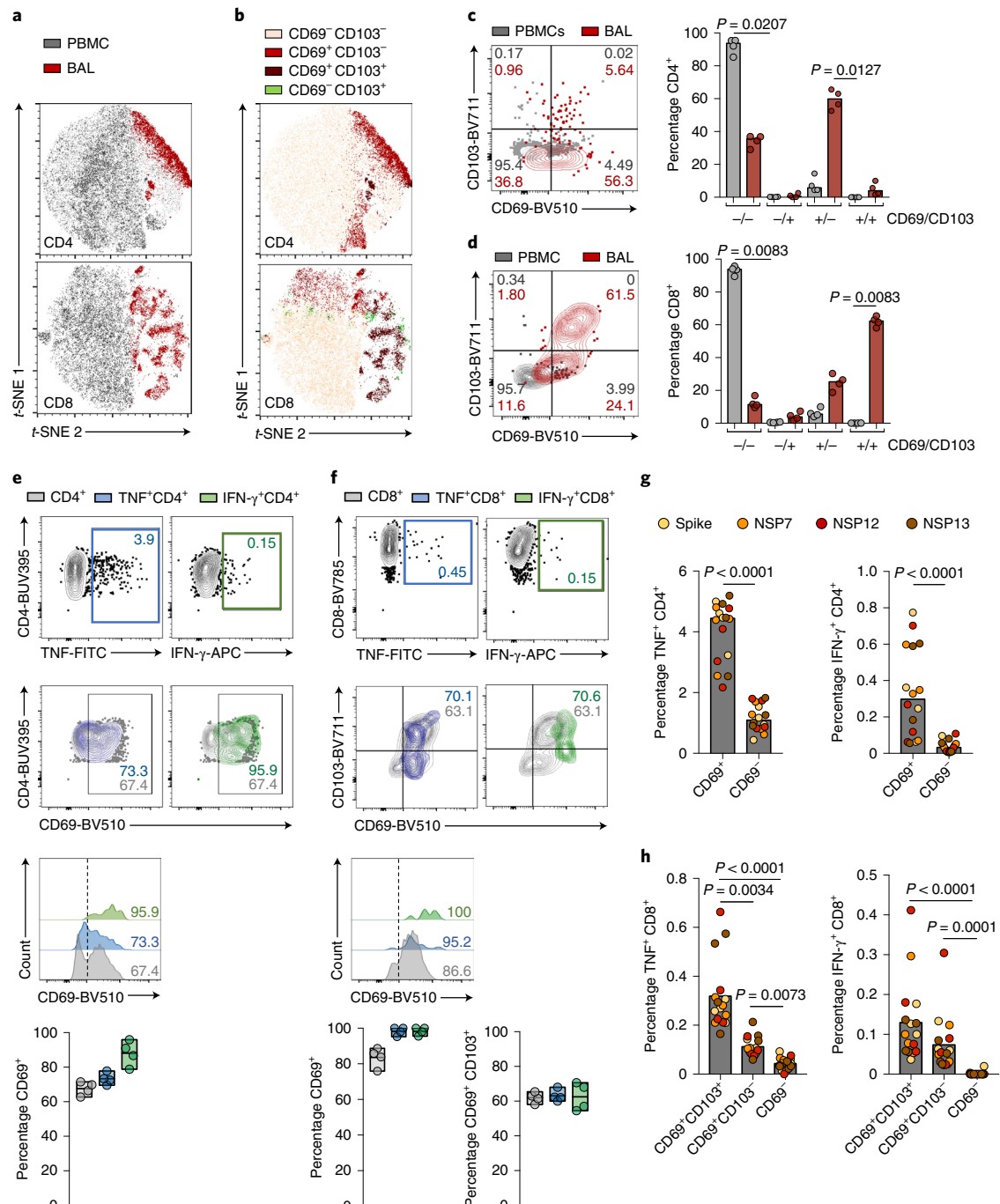

**Fig. 2 | Enrichment of resident phenotype in global and antigen-specific T cells in BAL. a**, The *t*-SNE plots of concatenated PBMCs and BAL CD4+ (top) or CD8+ T cells (bottom) highlighting sample tissue origin. **b**, The *t*-SNE plots of concatenated PBMCs and BAL CD4+ (top) or CD8+ T cells (bottom) highlighting expression of CD69 and CD103. **c**, Example plot and percentage of CD69 expression on CD4+ T cells from PBMCs or BAL. **d**, Example plot and percentage of CD103 and CD69 expression on CD8+ T cells from PBMCs or BAL. **e**, Example plots of TNF- or IFN-γ-producing CD4+ T cells (top) and CD69 expression on total, TNF+ or IFN-γ+CD4+ T cells (bottom) using NSP12 peptide pool. **f**, Example plots of TNF- or IFN-γ-producing CD8+ T cells (top) and CD103 versus CD69 expression on total, TNF+ or IFN-γ+ CD8+ T cells (bottom) using NSP12 peptide pool. **g**, Comparison of CD4+CD69+ and CD69− cells in BAL producing TNF (left) or IFN-γ (right) per peptide pool. **h**, Comparison of CD8+CD69+CD103+, CD69+CD103− and CD69− cells in BAL producing TNF (left) or IFN-γ (right) per peptide pool. **a–h**, Four biologically independent samples with the highest frequencies of IFN-γ production examined over one independent experiment. Bars at median (**c**, **d**, **g** and **h**); floating bars indicating the mean, minimum and maximum values within the dataset (**e** and **f**); Kruskal–Wallis one-way ANOVA test and Dunn's multiple comparison (**c–f** and **h**); Wilcoxon's paired test (**g**).

Immunoglobulin (Ig)G antibody titers against HCoVs were also higher in donors with SARS-CoV-2-reactive BAL T cells (significant for 229E, trends for NL63, OC43, HKU1; Fig. 4c), corroborating the evidence for more recent/stronger HCoV exposure

being associated with cross-reactive airway T cells. In contrast to the cross-reactive T cells we had observed, no antibodies able to crossreact with SARS-CoV-2 spike or nucleoprotein were detectable in these pre-pandemic sera (Fig. 4d).

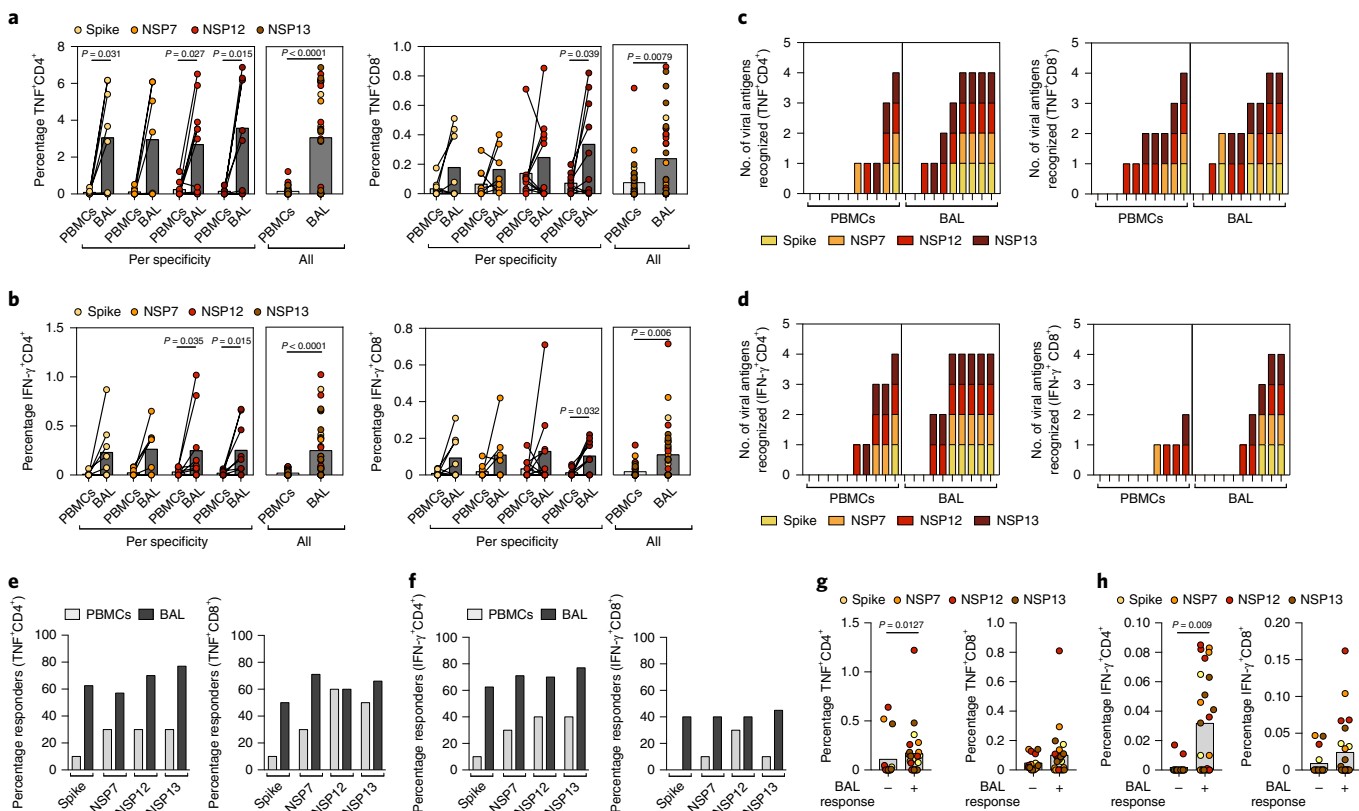

**Fig. 3 | SARS-CoV-2-specific T cells are enriched in the lungs. a,b,** Frequency of TNF- (**a**) or IFN-γ-producing (**b**) CD4+ and CD8+ T cells after overnight stimulation of BAL and PBMCs with Sars-CoV-2 peptide pools shown as paired samples for the different peptide pools separately (left box) or all combined (right box). **c,d,** Number of viral antigens recognized ranked by response level in TNF- (**c**) or IFN-γ-producing (**d**) CD4+ and CD8+ T cells. **e,f,** Percentage of responders in BAL or PBMC samples for TNF+CD4+ or CD8+ T cells (**e**) and IFN-γ+CD4+ or CD8+ T cells (**f**). Responses were considered positive when above mean + 2 s.d. of DMSO control after overnight stimulation with SARS-CoV-2 peptide pools. **g,h,** Percentage of TNF (**g**) or IFN-γ (**h**) production by CD4+ or CD8+ T cells after overnight stimulation of PBMCs distributed according to positive detection of responses in BAL (n = 10 biologically independent samples examined over one independent experiment). Bars at median (**a**, **b**, **g** and **h**); Wilcoxon's paired test (**a** and **b**); Mann–Whitney U-test (**g** and **h**).

A potential confounder for this cohort is the influenza vaccination and pneumococcal challenge that donors had received before BAL sampling, although these are 'real-world' physiological exposures, with >10% of the general population being pneumococcal carriers[14] and many receiving annual influenza vaccines. In most cases these had been administered >4 months before BAL; the two cases with a 6-week interval were not outliers in their SARS-CoV-2 reactivity (Extended Data Fig. 4a). We have previously noted that pneumococcal and influenza-specific memory responses account for ~1% of BAL T cells[15] (Extended Data Fig. 4b), whereas spontaneous T cell TNF release decays over time after pneumococcal challenge to become minimal >1 month post-challenge (Extended Data Fig. 4c), in line with the low levels of background cytokine production seen without peptide stimulation in this cohort. However, it remains possible that previous inflammatory exposures such as these increased the recruitment of SARS-CoV-2-reactive lung T$_{RM}$ cell specificities[16]. Previous challenge could also have activated global lung T cells, but BAL from a donor without pneumococcal challenge or influenza vaccine still had high expression of CD69 on CD4+ T cells, many of which co-expressed the additional T$_{RM}$ cell marker CD49a (Extended Data Fig. 4d).

We demonstrate that SARS-CoV-2-cross-reactive T cells can reside in the human airways, a key site for frontline protection against inhaled pathogens such as coronaviruses. SARS-CoV-2-reactive T cells were more frequent in the CD4+ than the CD8+ T cell fraction of pre-pandemic lower respiratory mucosa, but both

were capable of co-producing IFN-γ and TNF. Although this observational study could not examine their protective role, we speculate that airway memory T cells capable of such rapid, robust cytokine production would provide vital sentinel function, based on in vivo depletion studies in animal models. CD4+ T cell responses localized within the airway, rather than those in the lung parenchyma or vasculature, have been shown to be critical for protection against SARS-CoV; depletion of the airway fraction specifically abrogated the efficacy of mucosal vaccination[3]. Airway coronavirus-specific CD4+ T cells have been shown to mediate rapid IFN-γ-dependent induction of antiviral pathways such as STAT-1 (ref. [3]), consistent with our previous observation of healthcare workers aborting infection before PCR or antibody positivity[1]. The response by airway CD4+ T cells could include a cytotoxic component and might also initiate the production of CXCR3 chemokines to recruit migratory dendritic cells, able to amplify the CD8+ T cell response for final clearance of infected cells[3,17]; in line with this we found that frequencies of SARS-CoV-2-specific CD4+ and CD8+ T cells were correlated in donor BALs.

Six of ten of the healthy donors tested had T cells capable of cross-recognizing SARS-CoV-2 proteins within their airways before the SARS-CoV-2 pandemic. Although extrapolations cannot be made from such a small cohort to the wider population, enhanced protective immunity in this proportion is compatible with SARS-CoV-2 challenge and household exposure studies where around half of those exposed resisted infection[10,18–21]. T cells were reactive against

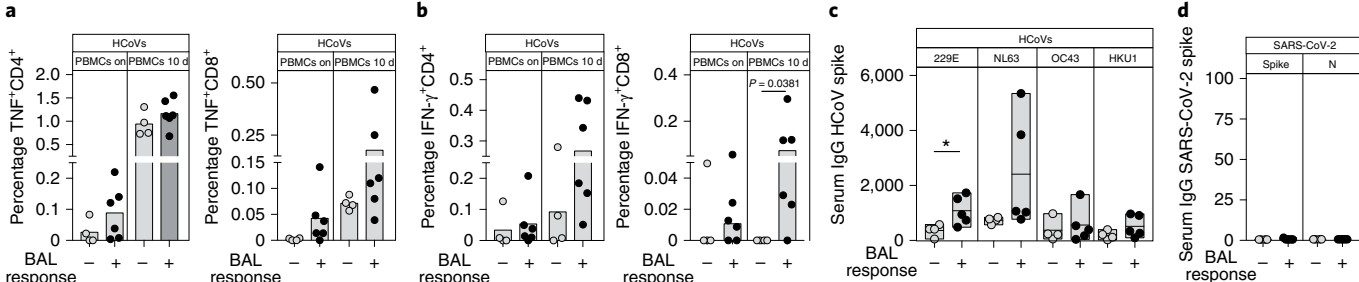

**Fig. 4 | HCoV spike-specific T cell and antibody responses are higher in individuals with detectable Sars-CoV-2-reactive T cell responses in BAL.**
**a,b**, Frequency of TNF- (**a**) or IFN-γ-producing (**b**) CD4$^+$ and CD8$^+$ T cells after overnight and 10-d stimulation of PBMCs with spike peptide pool from HCoVs 229E, NL63 and OC43 combined. **c**, Serum IgG to spike from HCoVs 229E, NL63, OC43 and HKU1 in arbitrary units ($n = 10$ (**a** and **b**) or $n = 9$ (**c** and **d**) biologically independent samples examined over one independent experiment). **d**, Serum IgG to Sars-CoV-2 spike and nucleoprotein in arbitrary units. **a,b**, Results are represented grouped as individuals who tested negative (BAL nonresponders) or positive (BAL responders) for Sars-CoV-2-specific T cells in BAL. Bars at median (**a** and **b**), floating bars indicating the mean, minimum and maximum values within the dataset (**c** and **d**); Mann–Whitney U-test (**a** and **d**).

all four regions of SARS-CoV-2 tested; critically, these included T cells specific for the RNA-dependent polymerase (NSP12) and other components of the core RTC (polymerase cofactor NSP7, helicase NSP13) of SARS-CoV-2, which we recently found associated with abortive seronegative infection. Thus, the responses we previously identified in the periphery were probably a representative subset of their more protective lung mucosa-resident counterparts, which either had not yet acquired tissue residence or were 'ex-T$_{RM}$ cells'[22,23]. Consistent with this, we noted that circulating polymerase (NSP12)-specific T cell frequencies correlated significantly with lung airway responses identified after overnight peptide re-stimulation. Pre-existing SARS-CoV-2-specific T cell frequencies were enriched in the BAL compared with the circulation. BAL cell yields did not allow a more comprehensive analysis of T cell specificities, but it is likely that responses targeting other regions of SARS-CoV-2 would be similarly enriched in the airways compared with blood. The frequency of lung T$_{RM}$ cells with cross-reactive potential against SARS-CoV-2 may also be underestimated because lavage samples only airway (epithelial) and not interstitial T$_{RM}$ cells, with the latter typically containing more T cells capable of cytotoxicity[24,25]. Although proliferating interstitial lung T$_{RM}$ cells can function as a reservoir of responses to seed and replenish the airway T$_{RM}$ cells[26], it is the airway T$_{RM}$ cells that have been found to form a critical component of protection against animal coronaviruses[3] and other respiratory viruses[24]. Maintaining a large memory pool of pathogen-specific T cells at the site of infection entry is a key protective advantage of the efficient frontline immunosurveillance provided by T$_{RM}$ cells in many tissues[12,27,28].

Lung T$_{RM}$ cells may not be as long-lived as other tissue T$_{RM}$ cells, although stable frequencies persisted for more than 1 year in some human lung transplant recipients[12,27,29,30]. We could not determine longevity because the timing and nature of the original infection priming the cross-reactive lung T cells that we observed are unknown. Despite the small size of the cohort, we were able to detect differences in the strength of T cell and humoral immunity to seasonal HCoVs in those with or without detectable SARS-CoV-2-cross-reactive responses in the airways. This suggests that the timing and/or dose of previous HCoV exposure may shape the effectiveness of airway T cell crossprotection against SARS-CoV-2. Although 'common cold viruses' do not typically cause lower respiratory symptoms, respiratory syncytial virus (RSV) has also recently been shown to induce T cells in the lower airways in healthy adults with only upper respiratory tract symptoms[13]. Pre-existing SARS-COV-2 T$_{RM}$ cells have been identified by activation assays in tonsillar lymphoid tissue extracted before the pandemic[31], suggesting that functional responses with

crossprotective potential may also develop in the upper airways. However, it is likely that some airborne virus can bypass immune defenses in the upper respiratory tract, underscoring the importance of the lower airway T cells, which we have identified, in preventing the lung pathology that causes severe infection outcomes. Future studies will need to investigate whether, in addition to the pre-pandemic cross-reactive responses observed, airway-resident functional memory T cells can be formed after SARS-CoV-2 infection. T cell receptor sequencing has provided evidence for SARS-CoV-2-specific CD8$^+$ T cells in nasal samples from four donors after severe COVID-19 (ref. [32]). Lung T$_{RM}$ cells have been shown to form either after SARS-CoV-2 infection in selected cases where tissue was obtained from surgical resection or post mortem[33,34], which did not allow airway responses to be distinguished from parenchymal/vascular T$_{RM}$ cells. It is not yet known whether airway or interstitial lung-resident T cell responses are induced by the current peripherally administered vaccines or whether mucosal delivery is required to achieve this.

Next-generation vaccines are being developed with greater focus on regions beyond the highly variable spike and on inducing mucosal immunity able to provide durable protection at the site of infection[35]. Our recent study highlighted the potential for RTC-specific T cells to abort early infection before PCR positivity and antibody seroconversion, potentially reducing transmission as well as disease and targeting early expressed viral replication proteins conserved across SARS-CoV-2 variants and other animal and HCoVs[1]. The data presented in the present study provide an explanation for the observed association between cross-reactive T cells and rapidly aborted infection, representing an immediate response at the site of infection after previous priming by, for example, an HCoV. Our findings underscore the rationale for vaccines delivered by the mucosal route, to expand tissue-resident T cells in the airways, including broadly cross-reactive SARS-CoV-2 RTC and spike specificities, aiming to provide frontline immunosurveillance against future variants.

### Online content

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

## Methods

**Study design and BAL collection.** This was a cross-sectional study, using residual samples left after two Experimental Human Pneumococcal Carriage (EHPC) model studies, which were conducted in Liverpool, UK before the COVID-19 pandemic. Briefly, ten healthy, nonsmoking, adults (aged 18–44 years), who had previously enrolled in two different EHPC studies from 2016 to 2018, underwent a one-off research bronchoscopy, as previously described[36]. Ten participants were challenged intranasally with live *S. pneumoniae* (serotype 6B) as previously described[37]. In addition, an influenza vaccine (live attenuated influenza vaccine (LAIV) or trivalent influenza vaccine (TIV)) was administered 3 d after pneumococcal challenge as part of one EHPC clinical trial in nine of the ten subjects in the present study (see Supplementary Table 1). BAL samples were obtained through research bronchoscopy between 1 month and 4 months post-pneumococcal challenge and influenza vaccine. Blood samples for sera and PBMC isolation were collected at the same day as BAL. One BAL sample obtained from an individual without previous pneumococcal challenge or influenza vaccination was used as the control.

**Sample processing.** BAL samples were processed as previously described[38], cryopreserved in CTL-CryoABC medium kit (Immunospot). After thawing, alveolar macrophages were routinely separated from other nonadherent immune cell populations using an adherence step, as previously described[36]. Blood was processed for sera collection or PBMCs were isolated from heparinized blood samples using density-gradient sedimentation layered over Ficoll-Paque in SepMate tube and then cryopreserved in CTL-CryoABC medium kit (Immunospot).

**Peptides.** A full list of the peptides contained in pools spanning the whole SARS-CoV-2 NSP7, NSP12 and NSP13 proteins (15-mer peptides overlapping by 10 amino acids) or spike (15-mer peptides based on predicted epitopes) has been previously described[1,39] (GL Biochem Shanghai Ltd, >80% purity). When insufficient BAL cells were available, pools were prioritized as follows: NSP12 ($n=10$) > NSP13 ($n=9$) > spike ($n=8$) > NSP7 ($n=7$). Assessment of crossreactivity in PBMCs was performed using peptide pools spanning spike protein from the endemic HCoVs OC43, 229E and NL63 combined based on equivalent, predominantly immunodominant regions of SARS-CoV-2 spike described above (BEI Resources, peptide arrays, 17-mers). The following reagents were obtained through BEI Resources, National Institute of Allergy and Infectious Diseases, National Institutes of Health: peptide array, HCoVs OC43, 229E, NL63 spike (S) glycoproteins, NR-53728, NR53727 and NR53729.

**Intracellular cytokine staining.** Mononuclear BAL cells (($0.8–1) \times 10^5$ cells per well) and PBMCs ($5 \times 10^5$ cells per well) were seeded in 96-well plates and stimulated with peptide pools (2 µg ml⁻¹ per peptide) in R10 supplemented with 0.5 µg ml⁻¹ of soluble anti-CD28, 20 U ml⁻¹ of recombinant human interleukin (IL)-2 and Brefeldin A (10 µg ml⁻¹; Sigma-Aldrich) for 16 h. For 10-d cultures, PMBCs were previously stained with CellTrace Violet cell proliferation kit following the manufacturer's instructions (Thermo Fisher Scientific) and seeded at $2 \times 10^5$ cells per well. On days 3 and 6, 100 µl of medium was removed and replaced with R10 supplemented with anti-CD28 and IL-2 as above. On day 9, PBMCs were re-stimulated with peptide pools (2 µg ml⁻¹ per peptide) and Brefeldin A (10 µg ml⁻¹; Sigma-Aldrich). After stimulation, PBMCs were harvested and stained for fixable live/dead (Near infrared, Thermo Fisher Scientific), followed by anti-human conjugated antibodies targeting surface proteins. After fixation and permeabilization (Cytofix/Cytoperm, BD Biosciences), cells were incubated with saturating concentrations of anti-human antibodies for intracellular staining. Antibodies used in the present study include: TNF FITC (BD Biosciences, clone MAb11; 1:50), CD8α BV785 (BioLegend, clone RPA-T8; 1:100), IFN-γ BV605 (BD Biosciences, clone B27; 1:100), IFN-γ antigen-presenting cell (APC) (BioLegend, clone 4S.B3; 1:50), CD3 BUV805 (BD Biosciences, clone UCHT1; 1:100), CD4 BUV395 or BV 421 (BD biosciences, clone SK3; 1:100), CD154 (CD40L) Pe-Cy7 (BioLegend, clone 24-31; 1:100), CD103 BV711 (BioLegend, clone ber-act8; 1:100), CD69 BV510 (BioLegend, clone fn50; 1:100) and CD49a BUV395 (BD Biosciences, clone SR84; 1:100). Samples were acquired on a BD LSRII flow cytometer using FACSDIVA v.9.0. FMOs (fluorescence minus one values) and unstimulated samples were used to determine gates applied across samples. Data were analyzed using FlowJo v.10.7 (TreeStar). An unstimulated control well was included for each sample and the percentage of cytokine producing CD4⁺ or CD8⁺ T cells was subtracted from all peptide stimulated wells. Reponses were considered positive when above dimethyl sulfoxide (DMSO) mean values + 2 s.d. BAL samples obtained for previous studies were stimulated with 1.2 µg ml⁻¹ of influenza antigens (TIV, 2016/2017) or *S. pneumoniae* for 16 h as previously described[15]. Cells were stained with Violet Viability dye (Thermo Fisher Scientific) and antibodies CD3 APC-H7 (clone SK7; 1:100), CD4 PerCP5.5 (clone SK3; 1:100), CD8 AF700 (clone SK1; 1:100), CD69 BV650 (clone FN5O; 1:100), CD103 BV605 (clone Ber-ACT8; 1:100), CD49a APC (clone TS2/7; 1:100),

IFN-γ PE (clone 4S.B3; 1:100) and TNF BV711 (clone MAb11; 1:100) (all from BioLegend) added.

**Antibody meso-scale discovery immunoassay.** A multiplexed meso-scale discovery immunoassay to measure IgG antibody responses to spike of Sars-CoV-2 and seasonal coronaviruses HKU1, OC43, 229E and NL63, and nucleoprotein of Sars-CoV-2 was performed as previously described[40]. Antibody concentration is presented in arbitrary units (AU) interpolated from the emitter-coupled logic signal of the internal standard sample using a four-parameter logistic curve fit.

**Viral qPCR.** Nucleic acids for viral quantitative PCR were extracted from one aliquot of 250 µl of oropharyngeal swab and/or 80–120 µl of nasosorption sample using the Purelink Viral RNA/DNA Mini Kit (Life Technologies Corp.) according to the manufacturer's instructions. We tested for a broad panel of respiratory viruses, including adenoviruses, parainfluenza viruses 1–4, human bocavirus, human coronaviruses OC43, NL63 and 229E, RSV (A and B), human metapneumovirus, human rhinoviruses, enteroviruses and human influenza viruses A42 and B43, as previously described[41].

**Statistics and reproducibility.** Data were assumed to have a non-Gaussian distribution and nonparametric tests were used throughout. For single-paired and unpaired comparisons Wilcoxon's matched-pairs, signed-rank test and a Mann–Whitney $U$-test were used, respectively. For multiple unpaired comparisons, Kruskal–Wallis one-way analysis of variance (ANOVA) with Dunn's correction was used. For correlations, Spearman's $r$ test was used. A $P < 0.05$ was considered significant. Prism v.7.0 and Excel v.16.16.09 were used for analysis. Details of the statistics are provided in the figure legends.

**Ethics statement.** All volunteers gave written informed consent and research was conducted in compliance with all relevant ethical regulations. Ethical approval was given by the North West National Health Service Research Ethics Committee (Ethics Committee reference nos. 14/NW/1460 and18/NW/0481, and Human Tissue Authority licensing no. 12548).

**Reporting summary.** Further information on research design is available in the Nature Research Reporting Summary linked to this article.

## Data availability

All data analyzed during the present study are included in this article and its supporting information files. Source data are provided with this paper.

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

## Acknowledgements

We thank all the patients and control volunteers who participated in the present study and all the clinical staff who helped with recruitment and sample collection. We thank A. Bertoletti for supplying the pre-pooled Sars-Cov-2 peptides. M.K.M. is supported by Wellcome Trust Investigator Award (no. 214191/Z/18/Z) and CRUK Immunology grant (no.26603), and L.S. by a Medical Research Foundation fellowship (044-0001). Collection of clinical samples was supported by the Bill and Melinda Gates Foundation (grant no. OPP1117728) and the UK Medical Research Council (grant no. M011569/1) awarded to D.M.F.

## Author contributions

M.O.D., E.M., D.M.F. and M.K.M. conceived the study. M.O.D., E.M., D.M.F., M.K.M. and L.S. designed the experiments. E.M., D.M.F. and J.R. obtained samples. E.M. processed the samples. M.O.D., E.M., M.J. and D.G. generated and analyzed the data.

M.O.D. and L.S set up the assays. M.O.D. and M.K.M. prepared the manuscript. All authors provided critical input into the manuscript.

## Competing interests

The authors declare no competing interests.

## Additional information

**Extended data** Extended data are available for this paper at https://doi.org/10.1038/s41590-022-01292-1.

**Correspondence and requests for materials** should be addressed to Daniela Ferreira or Mala K. Maini.

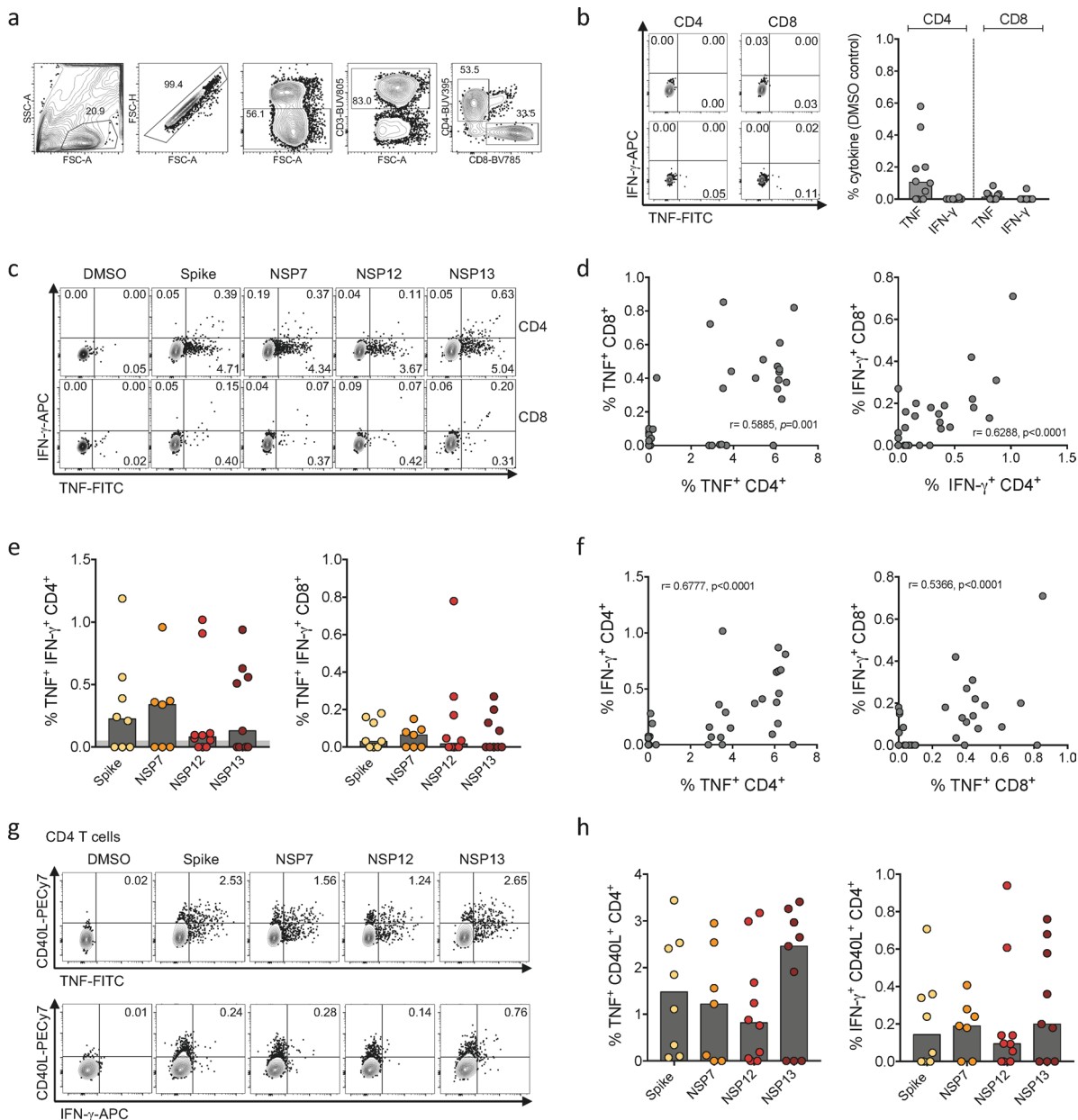

**Extended Data Fig. 1 | Detection of SARS-CoV-2-specific T cells in pre-pandemic BAL samples. a**, Example gating of intracellular cytokine staining of BAL after overnight peptide stimulation: Lymphocytes (SSC-A vs. FSC-A), single cells (FSC-H vs. FSC-A), Live cells (fixable live/dead-), CD3$^+$, CD4$^+$ or CD8$^+$. **b**, Example plot and frequencies of IFN-γ and TNF production by CD4$^+$ and CD8$^+$ T cells after overnight incubation of BAL samples with media containing DMSO (unstimulated control). **c**, Example plot of IFN-γ vs. TNF on CD4$^+$ and CD8$^+$ T cells after overnight stimulation of BAL samples with SARS-CoV-2 peptide pools. **d**, Correlation of TNF (left) or IFN-γ−producing (right) CD4$^+$ vs CD8$^+$ T cells in BAL. **e**, Frequency of CD4$^+$ and CD8$^+$ T cells co-expressing IFN-γ and TNF after overnight stimulation of BAL samples with SARS-CoV-2 peptide pools. **f**, Correlation of CD4$^+$ and CD8$^+$ T cells producing TNF vs IFN-γ in BAL. **g,h**, Example plots (**g**) of CD40L vs. TNF (top) or IFN-γ (bottom) gated on CD4$^+$ T cells and (**h**) percentage of CD4$^+$ T cells co-expressing CD40L and TNF or IFN-γ. n = 10 biologically independent samples examined over one independent experiment; **b,e,h**, bars at median, grey area represents mean + 2 SD of DMSO control; Kruskal-Wallis one-way ANOVA test and Dunn's multiple comparison; **d,f**, Spearman correlation.

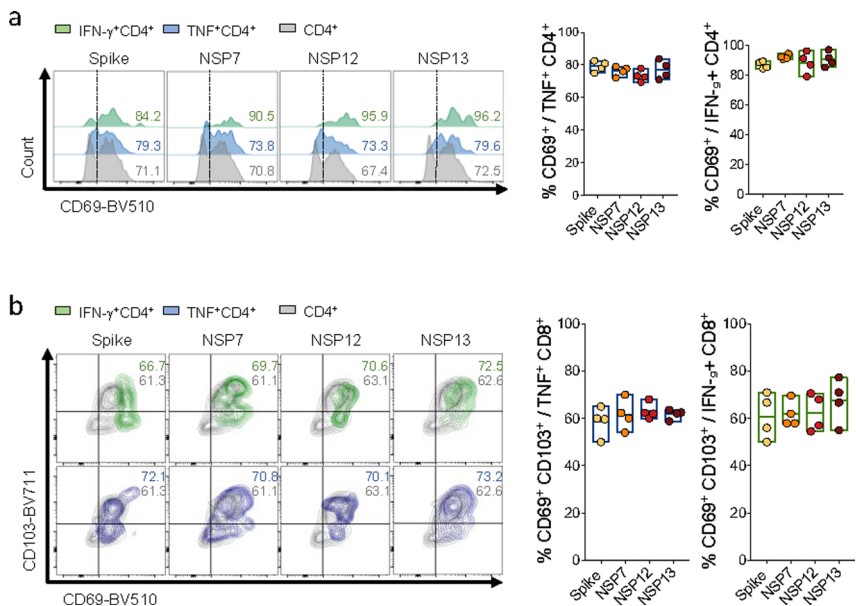

**Extended Data Fig. 2 | Enrichment of resident phenotype in global and antigen-specific T cells in BAL. a**, Representative histogram and percentages of CD69 expression on total, TNF+ or IFN-γ+ CD4+ T cells for the different peptide pools. **b**, Example plots and frequencies of CD103 vs. CD69 on total, TNF + or IFN-γ + CD8+ T cells for the different peptide pools. Analysis on 4 biologically independent samples with the highest frequencies of IFN-γ production examined over one independent experiment. Floating bars indicate the mean, minimum and maximum values within the dataset Kruskal-Wallis one-way ANOVA test and Dunn's multiple comparison.

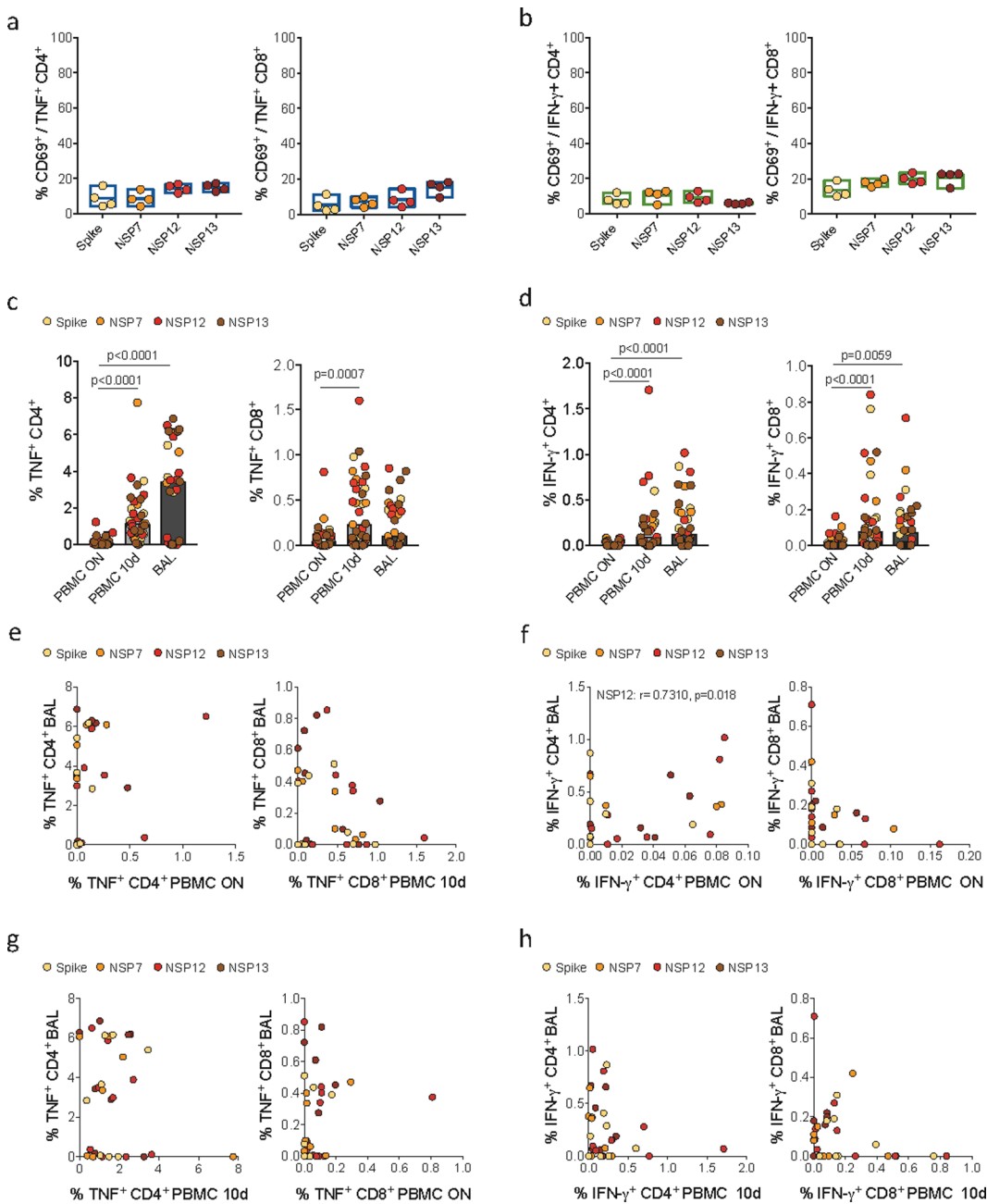

**Extended Data Fig. 3 | Lower frequencies of SARS-CoV-2-specific T cells are found in PBMC samples. a**, CD69 expression on TNF−producing CD4+ and CD8+ T cells in PBMCs. **b**, CD69 expression on IFN-γ−producing CD4+ and CD8+ T cells in PBMCs. **c,d**, Percentage of TNF (**c**) or IFN-γ (**d**) production by CD4+ or CD8+ T cells after overnight stimulation of BAL, or overnight (ON) or 10 days (10d) expansion of PBMCs with SARS-CoV-2 peptide pools. **e,f**, Correlation of TNF (**e**) or IFN-γ (**f**) responses for CD4+ or CD8+ T cells in BAL vs PBMC after overnight peptide stimulation (ON). **g,h**, Correlation of TNF (**g**) or IFN-γ (**h**) responses for CD4+ or CD8+ T cells in BAL vs PBMC after 10 days peptide expansion (10d). **a,b**, floating bars indicate the mean, minimum and maximum values within the dataset; **c,d** bars at median. **a,b**, analysis on 4 biologically independent PBMC samples from donors that showed the highest frequencies of IFN-γ production in BAL; **c-h**, n = 10 biologically independent samples examined over one independent experiment; **a-d**, Kruskal-Wallis one-way ANOVA test and Dunn's multiple comparison. **e-h**, Spearman correlation.

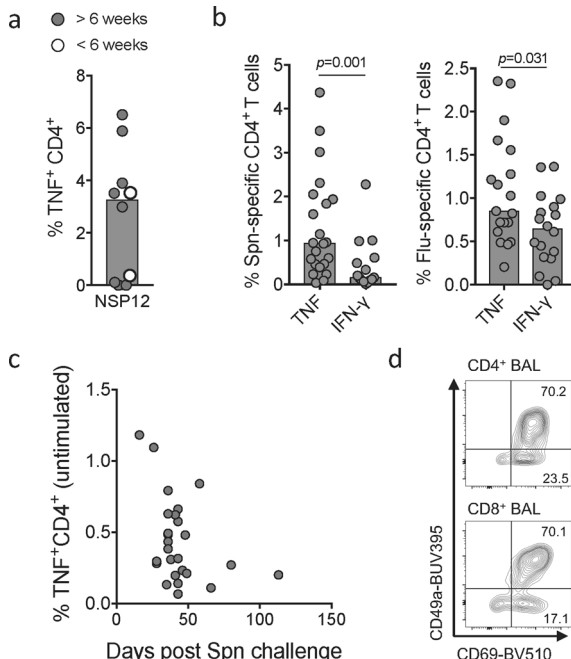

**Extended Data Fig. 4 | Low-level non-specific T cell responses in BAL more than 6 weeks after Spn challenge. a**, Frequency of TNF-producing CD4+ T cells after overnight stimulation of BAL samples with Sars-CoV-2 NSP12 peptide pool. White or gray dots show BAL samples collected earlier or later than 6 weeks after *Streptococcus pneumoniae* (Spn) challenge, respectively; n=10 biologically independent samples examined over one independent experiment **b**, Frequency of TNF or IFN-γ–producing CD4+ T cells after overnight stimulation of BAL samples with inactivated Spn (left) (n=22) or Flu virus (right) (n=19) in Spn challenged and vaccinated (LAIV) individuals. **c**, Frequency of TNF-producing CD4+ T cells after overnight incubation of BAL cells with DMSO using samples collected at different time points from 16-113 days after Spn Challenge (n=22). **d**, Example plot of CD4+ and CD8+ T cells from BAL collected from a unchallenged individual showing expression of CD49a *vs.* CD69 expression. **a-b**, bars at median. **b**, Mann-Whitney test; **c**, Spearman correlation.

# Reporting Summary

## Statistics

For all statistical analyses, confirm that the following items are present in the figure legend, table legend, main text, or Methods section.

| n/a | Confirmed | |
|---|---|---|
| ☐ | ☒ | The exact sample size (*n*) for each experimental group/condition, given as a discrete number and unit of measurement |
| ☐ | ☒ | A statement on whether measurements were taken from distinct samples or whether the same sample was measured repeatedly |
| ☐ | ☒ | The statistical test(s) used AND whether they are one- or two-sided *Only common tests should be described solely by name; describe more complex techniques in the Methods section.* |
| ☒ | ☐ | A description of all covariates tested |
| ☐ | ☒ | A description of any assumptions or corrections, such as tests of normality and adjustment for multiple comparisons |
| ☐ | ☒ | A full description of the statistical parameters including central tendency (e.g. means) or other basic estimates (e.g. regression coefficient) AND variation (e.g. standard deviation) or associated estimates of uncertainty (e.g. confidence intervals) |
| ☐ | ☒ | For null hypothesis testing, the test statistic (e.g. *F*, *t*, *r*) with confidence intervals, effect sizes, degrees of freedom and *P* value noted *Give P values as exact values whenever suitable.* |
| ☒ | ☐ | For Bayesian analysis, information on the choice of priors and Markov chain Monte Carlo settings |
| ☒ | ☐ | For hierarchical and complex designs, identification of the appropriate level for tests and full reporting of outcomes |
| ☒ | ☐ | Estimates of effect sizes (e.g. Cohen's *d*, Pearson's *r*), indicating how they were calculated |

*Our web collection on statistics for biologists contains articles on many of the points above.*

## Software and code

Policy information about availability of computer code

| Data collection | No software was used to collect data. |
|---|---|
| Data analysis | Software used for data/statistical analysis: FlowJo v.10.7.1; FACSDIVA v9.0; Prism 7.0; Excel v.16.16.09; |

For manuscripts utilizing custom algorithms or software that are central to the research but not yet described in published literature, software must be made available to editors and reviewers. We strongly encourage code deposition in a community repository (e.g. GitHub). See the Nature Portfolio guidelines for submitting code & software for further information.

## Data

Policy information about availability of data

All manuscripts must include a data availability statement. This statement should provide the following information, where applicable:

- Accession codes, unique identifiers, or web links for publicly available datasets
- A description of any restrictions on data availability
- For clinical datasets or third party data, please ensure that the statement adheres to our policy

All data analysed during this study are included in this published article and its supporting information files.

# Human research participants

Policy information about studies involving human research participants and Sex and Gender in Research.

| | |
|---|---|
| Reporting on sex and gender | Five female and five male donors participated in this study, detailed information is provided in Extended Data Table 1. |
| Population characteristics | Population characteristics and treatments have are described in Methods and Supplementary materials. Ten healthy, non-smoking, adults (aged 18 - 44 years), were enrolled in two different EHPC studies from 2016 to 2018. Participants were challenged intranasally with live Streptococcus pneumoniae (serotype 6B) and immunised with influenza vaccine (LAIV or TIV) where applicable. BAL samples were obtained through research bronchoscopy between 1 and 4 months post pneumococcal challenge. |
| Recruitment | All volunteers gave written informed consent. Adult (>18 years) were invited to participate via email and posters. Key eligibility criteria included capacity to give informed consent, no immunocompromised state or contact with susceptible individuals, no pneumococcal or influenza vaccine or infection in the last 2 years and not having taken part in EHPC studies in the past 3 years. |
| Ethics oversight | North West National Health Service Research Ethics Committee (Ethics Committee reference numbers: 14/NW/1460 and 18/NW/0481, and Human Tissue Authority licensing number: 12548 |

Note that full information on the approval of the study protocol must also be provided in the manuscript.

# Field-specific reporting

Please select the one below that is the best fit for your research. If you are not sure, read the appropriate sections before making your selection.

☒ Life sciences ☐ Behavioural & social sciences ☐ Ecological, evolutionary & environmental sciences

For a reference copy of the document with all sections, see nature.com/documents/nr-reporting-summary-flat.pdf

# Life sciences study design

All studies must disclose on these points even when the disclosure is negative.

| | |
|---|---|
| Sample size | Sample sizes are given for each figure throughout the paper when individual dots are not shown. Sample size can vary across figure panels depending on which stimulations were performed (limited by number of BAL cells recovered). |
| Data exclusions | No participant or individual samples were excluded after data was generated. |
| Replication | Due to limited sample availability experiments were not replicated. |
| Randomization | To fit the aim of the study, all samples were considered as from one group: individuals unexposed to Sars-CoV-2 (pre-pandemic). Experiments were performed with protocols optimised to reduce batch variation and to ensure mixing of experimental groups across batches e.g Flow cytometer parameters were consistent between runs (No MFI comparisons were performed, only gating and percentage of parent). |
| Blinding | Experiments were not randomized and the investigators were not blinded to allocation during experiments and outcome assessment since all samples belonged to a single untreated group. However, experimental set-up and controls ensured accurate analysis and interpretation. |

# Reporting for specific materials, systems and methods

We require information from authors about some types of materials, experimental systems and methods used in many studies. Here, indicate whether each material, system or method listed is relevant to your study. If you are not sure if a list item applies to your research, read the appropriate section before selecting a response.

## Materials & experimental systems

| n/a | Involved in the study |
|---|---|
| ☐ | ☒ Antibodies |
| ☒ | ☐ Eukaryotic cell lines |
| ☒ | ☐ Palaeontology and archaeology |
| ☒ | ☐ Animals and other organisms |
| ☐ | ☒ Clinical data |
| ☒ | ☐ Dual use research of concern |

## Methods

| n/a | Involved in the study |
|---|---|
| ☒ | ☐ ChIP-seq |
| ☐ | ☒ Flow cytometry |
| ☒ | ☐ MRI-based neuroimaging |

# Antibodies

| | |
|---|---|
| Antibodies used | Detailed information regarding all antibodies used in this study are listed in the methods with manufacturer, clone, and dilution used. Antibodies used in this study: TNF FITC (BD bioscience, clone MAb11; 1:50), CD8α BV785 (Biolegend, clone RPA-T8; 1:100), IFN-g BV605 (BD biosciences, clone B27; 1:100), IFN-g APC (Biolegend, clone 4S.B3; 1:50), CD3 BUV805 (BD biosciences, clone UCHT1; 1:100), CD4 BUV395 or BV 421 (BD biosciences, clone SK3; 1:100), CD154 (CD40L) Pe-Cy7 (Biolegend, clone 24-31; 1:100), CD103 BV711 (Biolegend, clone ber-act8; 1:100), CD69 BV510 (Biolegend, clone fn50; 1:100), CD49a BUV395 (BD biosciences, clone SR84; 1:100), CD3 APC-H7 (Biolegend, clone SK7; 1:100), CD4 PerCP5.5 (Biolegend, clone SK3; 1:100), CD8 AF700 (Biolegend, clone SK1; 1:100), CD69 BV650 (Biolegend, clone FN5O; 1:100), CD103 BV605 (Biolegend, clone Ber-ACT8; 1:100), CD49a APC (Biolegend, clone TS2/7; 1:100), IFN-g PE (Biolegend, clone 4S.B3; 1:100), and TNF BV711 (Biolegend, clone MAb11; 1:100). |
| Validation | All antibodies were purchased from well established manufacturers and were validated by the vendor for species and target. e.g. BD biosciences, Biolegend in Knock-out/knock-in primary model systems to ensure biological accuracy in ISO 9001 certified facilities. Side-by-side lot comparisons are performed. Details of antibody clones have been included for cross-referencing of manufacturing company specification/validation processes. We further validated antibodies by titration to optimal concentrations and by using positive controls where possible (e.g. using populations known to express a certain marker or by polyclonal stimulation). Fluorescence minus one stains or unstimulated wells were used to define gates in Flowjo for all FACS assays. Negative controls were included in each run and positive controls where possible (PBMCs). All data is presented as background subtracted as described in the methods. |

# Clinical data

Policy information about clinical studies

All manuscripts should comply with the ICMJE guidelines for publication of clinical research and a completed CONSORT checklist must be included with all submissions.

| | |
|---|---|
| Clinical trial registration | n/a |
| Study protocol | North West National Health Service Research Ethics Committee reference numbers: 14/NW/1460 and18/NW/0481. Human Tissue Authority licensing number: 12548. |
| Data collection | Data collection was conducted in the Liverpool School of Tropical Medicine (Liverpool, UK) from 2016 to 2018. |
| Outcomes | n/a |

# Flow Cytometry

## Plots

Confirm that:

☒ The axis labels state the marker and fluorochrome used (e.g. CD4-FITC).

☒ The axis scales are clearly visible. Include numbers along axes only for bottom left plot of group (a 'group' is an analysis of identical markers).

☒ All plots are contour plots with outliers or pseudocolor plots.

☒ A numerical value for number of cells or percentage (with statistics) is provided.

## Methodology

| | |
|---|---|
| Sample preparation | Detailed sample preparation is given in methods. All FACS was performed on frozen and thawed PBMC or BAL isolated by density gradient separation. Peripheral blood mononuclear cells (PBMC) were isolated from heparinized blood samples using Pancoll (Pan Biotech) or Histopaque®-1077 Hybri-MaxTM (Sigma-Aldrich) density gradient centrifugation in SepMate tubes (StemCell) according to the manufacturer's specifications. |
| Instrument | BD biosciences LSRII and Fortessa-X20 flow cytometers. |
| Software | FACS DIVA version 9.0 was used on instrument and exporting .fcs files were analysed in FlowJo version 10.7.1 (TreeStar). |
| Cell population abundance | BAL was stimulated using non-adherent cells after 4 hours resting at 37oC. PBMC and BAL were stained and run without sorting or enrichment. |
| Gating strategy | Example gating strategy and plots are given in Extended Data Figure 1. Data is reported as a percentage of lymphocytes/singlets/live/CD3+/CD4+ or CD8+ defining antigen specificity by production of IFNg, TNFa, CD40L or combinations of those. |

☒ Tick this box to confirm that a figure exemplifying the gating strategy is provided in the Supplementary Information.

