## [Peer Review File · Nature Immunology]

Peer Review Information

Journal: Nature Immunology

Manuscript Title: Airway-resident T cells from unexposed individuals cross-recognise SARS-CoV-2

Corresponding author name(s): Daniela Ferreira, Mala K Maini

Reviewer Comments & Decisions:

Decision Letter, initial version:
--

Subject: Decision on Nature Immunology submission NI-LE33921

Message: 20th May 2022

Dear Professor Maini,

As you know your Letter, "Airway-resident functional T cell responses against SARS-CoV-2 in unexposed individuals" has now been seen by 3 referees. You will see from their comments below that while they find your work of interest, some important points are raised. We are very interested in the possibility of publishing your study in Nature Immunology, but would like to consider your response to these concerns in the form of a revised manuscript before we make a final decision on publication.

Please revise your letter as outlined in recent response to the referee comments, In particular, we would welcome that you check for T cells responding to HCoV in remaining paired PBMC from this cohort to see whether these are increased in the 6 donors who had BAL T cells cross-reacting to SARS-CoV-2.

We therefore invite you to revise your manuscript taking into account all reviewer and editor comments. Please highlight all changes in the manuscript text file in Microsoft Word format.

* If you have not done so already please begin to revise your manuscript so that it conforms to our Letter format instructions at <http://www.nature.com/ni/authors/index.html>. Refer also to any guidelines provided in this letter.

* Please include a revised version of any required reporting checklist. It will be available to referees to aid in their evaluation of the manuscript goes back for peer review. They are available here:

Reporting summary:

Please use the link below to submit your revised manuscript and related files:
[REDACTED]

We hope to receive your revised manuscript within two weeks. If you cannot send it within this time, please let us know. We will be happy to consider your revision so long as nothing similar has been accepted for publication at Nature Immunology or published elsewhere.

Nature Immunology is committed to improving transparency in authorship. As part of our efforts in this direction, we are now requesting that all authors identified as 'corresponding author' on published papers create and link their Open Researcher and Contributor Identifier (ORCID) with their account on the Manuscript Tracking System (MTS), prior to acceptance. ORCID helps the scientific community achieve unambiguous attribution of all

scholarly contributions. You can create and link your ORCID from the home page of the MTS by clicking on 'Modify my Springer Nature account'. For more information please visit www.springernature.com/orcid.

Sincerely,

Jamie D.K. Wilson, D.Phil
Chief Editor
Nature Immunology
212 726 9207
j.wilson@us.nature.com

Reviewers' Comments:

Reviewer #1:

Remarks to the Author:

This manuscript by Diniz reported the detection of SARS-CoV-2 specific T cell responses in BAL samples obtained from pre-pandemic healthy volunteers, after receiving live pneumococci (intranasally challenged) as well as influenza vaccine (LAIV or TIV) as part of EHPC clinical trial. Intracellular cytokine staining of TNF alpha and IFN gamma post peptide stimulation were the main read out for T cell responses. While I found the topic is important, sample collection is unique/valuable, detection of SARS-CoV-2 responses in BAL is interesting with potential important implication regarding pre-existing local immunity to SARS-CoV-2. I feel the data presented in the current manuscript is too preliminary, the importance/functional relevance of such responses in protection is not clear. In addition, the samples were taken after live intranasal pneumococci challenge and influenza vaccination, both presumably would affect the statuses of the cell activation in lower respiratory airway, hence the immunophenotyping i.e., CD69 expression on T cells, spontaneous cytokine release etc.

Specifically:

1. I am not sure if it is appropriate to use CD69 as marker for tissue residence in this particular experimental setting (Figure 2), to distinguish activated T cell vs Trm cells, because CD69 is known as a surrogate marker of T-cell responsiveness to mitogen and Ag stimulus and often used as a measure of T-lymphocyte activation. To me it is therefore not surprising to see the correlation with CD69+ and antigen specificity post peptide stimulation (fig 2g and H).

2. Figure 1C: I am surprised to see that in some individuals with very high frequency (>20%) of CD4 T cells in BAL are SARS-CoV-2 specific, keeping in mind that those individuals just had recent intranasal live pneumococci challenge as well as influenza vaccination, would author expecting more of specific CD4 responses to those newly encounter antigens rather than SARS-CoV-2 specific T cells? It would be important to show the raw Facs data/background staining of each individual (i.e isotope control) in particular those with high frequency responses detected, this also applies to Figure 2F.

3. Figure 4, the data presented is a bit too preliminary in my view, although I agree it is important to develop method exploring the potential use of circulating T cell to represent what's going on in lung. However, just measuring the magnitude of the responses after short term culture is unlikely to be able to address such complicated issue properly - due to too many co-founding factors, for example, despite the possibility of biased expansion, other key factors such as the differences in epitope specificity, pattern of the responses as well as the composition of SARS-CoV-2 specific T cell repertoire also need to be considered.

4. Only TNF alpha and IFN gamma production are measured (although IL-2 and MIP-1beta were mentioned in the method, but no data were shown/discussed). The functional relevance and correlates with protection of those T cells and level of cross reactivity should be discussed, in particular in comparison with immune responses induced by nature infection.

Reviewer #2:

Remarks to the Author:

In this study, the authors hypothesize that pre-existing cross-reactive T cells recognizing the SARS-CoV-2 RTC (and have been previously correlated with early viral control) reside in human airways. To answer their question of whether cross-reactive T cells are present in human airways, the authors analyzed pre-pandemic BAL samples (along with the matched PBMC samples) of 10 subjects, and detected SARS-CoV-2 RTC-reactive T cells in the BAL of more than half of the subjects, with CD4 T cells being the predominant T cell subset in the airways. This study provides an important insight into the airway pre-existing resident T cell populations in human subjects, but the extent to which these T cells are protective or what their role is in prevention of infection remains unclear.

Specific comments:

1. In the conclusion section of the article, the authors write "The data presented here provide an explanation for the observed association between polymerase-cross-reactive T cells and rapidly aborted infection, representing an innate-like immediate response at the site of infection and potentially amplifying subsequent memory responses".

While the existence of cross-reactive TRM in the airways is a potential explanation for abortive infections, no evidence was shown in this study regarding the innate-like function of these T cells, as well as amplification of memory responses following reactivation. Besides the production of IFN γ and TNF α , could the authors comment on how does the activation of these T cells result in protection against virus?

2. Related to point 1), it is of interest that the frequency of antigen-specific CD4 T cells in the lung is considerably greater than that of antigen-specific CD8 T cells (Figure 1). In fact, the total SARS-CoV-2 specific CD4 T cell response seems remarkably high (10% in Fig 1c)! How do the authors believe that these CD4 T cells contribute to control of virus? Is it through T cell help and rapid induction of nAbs? Or is it through some other mechanism?

3. A key premise of the paper is that the SARS-CoV-2 specific T cells in the unexposed individuals are in fact cross-reactive T cells from prior exposure to other coronaviruses, presumably the common cold viruses. Could the authors provide some evidence for this? If mononuclear cells from BAL are stimulated with overlapping peptides from the spike proteins or RTC from the major common cold viruses, what would the frequencies of antigen-specific CD4 and CD8 T cells be?

4. Lung TRM have been shown in several mouse infection models to wane over time (for e.g. in [1]) and several other studies highlight the impermanence of the lung TRM compartment [2]. Given the potential waning of this cell population in the airways, what do the authors think in terms of the potential of lung TRM to induce long-lasting protection against respiratory infections?

5. How representative is this 10-subject cohort of the general population? Could their previous challenge with live pneumococci (or the LAIV) have induced an environment in the lung that recruits more T cells in a non-specific manner simply due to the inflammation present in the lungs? Previous studies have shown lung TRM recruitment without the need for specific antigen when there is inflammation [3].

[1] Slutter, B. et al. Dynamics of influenza-induced lung-resident memory T cells underlie waning heterosubtypic immunity. *Sci. Immunol.* 2, eaag2031 (2017)

[2] Zheng, M.Z.M., Wakim, L.M. Tissue resident memory T cells in the respiratory tract. *Mucosal Immunol* 15, 379–388 (2022). [https://doi-org.stanford.idm.oclc.org/10.1038/s41385-021-00461-z](https://doi.org/10.1038/s41385-021-00461-z)

[3] Chung H, Kim EA, Chang J. A “Prime and Deploy” Strategy for Universal Influenza Vaccine Targeting Nucleoprotein Induces Lung-Resident Memory CD8 T cells. *Immune Netw.* 2021 Aug;21(4):e28. <https://doi.org/10.4110/in.2021.21.e28>

Reviewer #3:

Remarks to the Author:

A. This is an interesting and potentially important report that describes an assessment of T cell response to coronavirus peptides in lower airway T cells that were frozen prior to the SARS-CoV-2 pandemic. The team report large populations of cells that respond to these peptides and have features of tissue resident memory populations.

B. This is a novel and original report.

C. The quality of the presentation and writing is very strong. The experimental methodology is well described.

D. Appropriate. The work is not overstated and the limitation of the small sample size is clearly referenced.

E. The work is robust. The work relies on the specificity of the intracellular cytokine analysis after overnight peptide stimulation. The responses are clearly above DMSO background. It might be suggested that the use of a peptide pool from a heterologous antigen might have been useful in order to assess potential impact of non-specific activation but cell number is limited and the work is now complete. The increase from tissue cells compared to blood appears robust

F. I do not think that further experimental work is needed and assessment of these sample cohort is complete. Limitations in relation to interpretation reflect the small sample size.

The CD69 expression on peripheral blood lymphocytes is high at 7.6% of CD4+ cells and beyond what is normally considered as a recently activated population within peripheral blood. Do the authors have any comments on this ?

I am not sure if the title is quite right as it suggests the cells have been primed to target SARS-CoV-2 – “Airway-resident functional T cell responses against SARS-CoV-2 in

unexposed individuals". Perhaps "Airway-resident T cells with cross-reactive recognition of SARS-CoV-2 in unexposed individuals".

Minor typo in Figure 3 where (b) is labelled as (c).

G. Appropriate

H. Very clear

Author Rebuttal to Initial comments

Manuscript number: NI-LE33921, Point-by-point response.

Many thanks for the opportunity to resubmit our letter by *Diniz, Mitsi et al* to Nature Immunology. We are pleased to read that all three reviewers appreciated the novelty of our human lower airway samples, and the interest and importance to the field of our findings. We have now addressed all the reviewers' comments and hope you agree that the manuscript is considerably strengthened as a result. In particular, we now include some striking new data showing that those donors with T cells cross-reactive against SARS-CoV-2 in their BAL had stronger immunity to seasonal coronaviruses. David Goldblatt and Marina Johnson have been added as co-authors for their work analysing seasonal coronavirus serology. Our responses to each of the reviewers' specific concerns are addressed in our point-by-point response below and all additions are underlined in the manuscript text.

Reviewer #1:**Remarks to the Author:**

... While I found the topic is important, sample collection is unique/valuable, detection of SARS-CoV-2 responses in BAL is interesting with potential important implication regarding pre-existing local immunity to SARS-CoV-2. I feel the data presented in the current manuscript is too preliminary, the importance/functional relevance of such responses in protection is not clear. In addition, the samples were taken after live intranasal pneumococci challenge and influenza vaccination, both presumably would affect the status of the cell activation in lower respiratory airway, hence the immunophenotyping i.e., CD69 expression on T cells, spontaneous cytokine release etc.

We thank the reviewer for their recognition of the value and interest of our study. We have extended our analyses to now show an important link between airway T cells able to cross-react with SARS-CoV-2 and the strength of immunity to human seasonal coronaviruses (HCoV). However we agree that the protective potential of such responses is difficult to determine in human studies. Very few of the large number of high-impact COVID-19 immunology studies to date have even attempted to assess correlates of protection in humans. However, we have shown an association of polymerase-specific T cells with abortive infection in the absence of antibodies (*Swadling et al Nature 2022*); crucially, we now show their presence in pre-pandemic healthy airway samples, the site at which they are thought to act to mediate such rapid shut-down of infection. In fact, despite the small cohort in this manuscript, frequencies of peripheral polymerase-specific T cells correlated with those within BAL. Functional protection cannot be attributed to immune responses in a particular compartment in human studies; even future comparison of protection from mucosal versus peripheral T cell vaccines will not be able to dissect the contribution of upper versus lower airway or localised lymphoid tissue responses. Fortunately, elegant work from the Perlman group has already demonstrated the critical contribution of the responses we identified here, namely CD4 T cells localised in the lower airways (using specific depletion of this component only) for protection against coronaviruses (*Zhao et al Immunity 2016*). These points have been added to the discussion (Line 188).

We have now emphasised that the prior live intranasal pneumococcal challenge and influenza vaccine were not recent, being more than 4 months previously in all but 2 subjects (the latter 2, who were 6 weeks post-challenge, were not outliers for any T cell assessments). We also now point out that these are 'real-world physiological' exposures since >10% of the general population are pneumococcal carriers at any one time and many receive annual 'flu' vaccines (Line 166).

Specifically:

1. I am not sure if it is appropriate to use CD69 as marker for tissue residence in this particular experimental setting (Figure 2), to distinguish activated T cell vs Trm cells, because CD69 is known as a surrogate marker of T-cell responsiveness to mitogen and Ag stimulus and often used as a measure of T-lymphocyte activation. To me it is therefore not surprising to see the correlation with CD69+ and antigen specificity post peptide stimulation (fig 2g and H).

Our key finding is the presence of CoV-2-reactive T cells in airways and the fact that these are greatly enriched compared to blood is the best evidence we have for a proportion being tissue resident; the phenotypic assessment of tissue-resident memory T cell (T_{RM}) status is of secondary importance. We agree that CD69 has to be interpreted cautiously as a T_{RM} phenotype rather than activation marker when T cells have been antigen-stimulated; we therefore avoided using it as an 'AIM' marker to detect antigen-specific T cells in this study. Overnight stimulation did not result in high expression of CD69 amongst peptide-responsive populations in the periphery (new Extended Data Fig. 3a). This contrasted with the high CD69 expression of peptide-specific T cells described above in the airway, reinforcing the latter being a feature of T_{RM} rather than simply a result of peptide activation. We have now also shown that a large proportion of unstimulated global airway CD4 express CD69, confirming it is a valid tissue-residency marker. We were able to use co-expression of CD69 and CD103 to define CD8 T_{RM} , but airway CD4s in some donors expressed insufficient CD103 to rely on this. We have further discussed the limitations of our T_{RM} phenotyping of SARS-CoV-2-specific CD4 T cells based on CD69 alone, which would not affect our main conclusions (Line 178). We have included new data (from a donor without prior pneumococcal challenge or influenza vaccination) showing a large proportion of unstimulated CD69-expressing airway CD4 co-express the T_{RM} marker CD49a, which is not an activation marker (new Extended Data Fig.4d).

2. Figure 1C: I am surprised to see that in some individuals with very high frequency (>20%) of CD4 T cells in BAL are SARS-CoV-2 specific, keeping in mind that those individuals just had recent intranasal live pneumococci challenge as well as influenza vaccination, would author expecting more of specific CD4 responses to those newly encounter antigens rather than SARS-CoV-2 specific T cells? It would be important to show the raw Facs data/background staining of each individual (i.e isotope control) in particular those with high frequency responses detected, this also applies to Figure 2F.

We have provided raw data to confirm the frequency and specificity of CD4 responses cross-recognising SARS-CoV-2 and showing the background from unstimulated control wells that was subtracted for all donors, in addition to using appropriate FMOs for flow gating (new Extended Data Fig. 1b). TNF-producing CD4 T cells have been found to dominate the response in previous BAL studies of RSV (*Guvanel et al, JCI 2019*) and the peripheral T cell response to SARS-CoV-2 has been noted to be CD4-dominated in multiple studies.

We have now further clarified that the pneumococcal and influenza administration were actually administered more than 4 months prior to BAL in the majority of donors, when any resulting inflammation would have resolved; the 2 subjects who were only 6 weeks post-challenge were not outliers in terms of responses measured (new Extended Data Fig.4a). We have added our previous unpublished data and referred to our published data showing that pneumococcal and flu-specific T cells have mean frequencies of ~1% of CD4 in BAL around this time point post-challenge/vaccination; these responses were to whole pneumococcus or influenza in a single well, requiring *in vitro* antigen processing and presentation and therefore expected to be lower than CoV-2-reactive T cells assessed by pools of peptides in 4 wells. Reassuringly, we found that the proportion of CD4 T cells producing TNF correlated closely with the proportion co-producing IFN-g +/- expressing CD40L (new Fig. 1e). However we have now discussed the possibility that antigen-specific CD4 T cell production of TNF could have been further amplified by secondary bystander responses (*Ge et al, Cell Reports 2019*) (Line 80); in order to avoid this potential partial bystander effect being inappropriately magnified, we have now avoided summing together the responses, instead showing cytokine responses to individual peptide pools separately in all graphs. Previous pneumococcal challenge and flu vaccine could have affected spontaneous cytokine release, but this was adjusted for by subtracting background from unstimulated wells for all SARS-CoV-2-specific analysis. We have added our previous data revealing that spontaneous TNF production from BAL CD4 T cells decays over time to become minimal >1month post-challenge, in line with the low levels (<0.6% of unstimulated T cells) seen in this cohort (new Extended Data Fig.4c).

3. Figure 4, the data presented is a bit too preliminary in my view, although I agree it is important to develop method exploring the potential use of circulating T cell to represent what's going on in lung. However, just measuring the magnitude of the responses after short term culture is unlikely to be able to address such complicated issue properly - due to too many co-founding factors, for example, despite the possibility of biased

expansion, other key factors such as the differences in epitope specificity, pattern of the responses as well as the composition of SARS-CoV-2 specific T cell repertoire also need to be considered.

We agree that our peripheral versus BAL comparison is limited in scope due to constraints of sample availability so have now given these data less weight in the revised manuscript, moving them into supplementary data figures to make space for the more compelling new HCoV data (see response to Reviewer 2, point 3).

4. Only TNF alpha and IFN gamma production are measured (although IL-2 and MIP-1beta were mentioned in the method, but no data were shown/discussed). The functional relevance and correlates with protection of those T cells and level of cross reactivity should be discussed, in particular in comparison with immune responses induced by nature infection.

We apologise that we left IL-2 and MIP1b in the methods and have now removed these since their measurement was uninformative in this study. If the reviewer/editor prefers, we could add the co-staining with IL-2 and MIP-1b as extended data but these were only detected at low levels and did not add to the existing multifunctional data we present with TNF, IFN-g and CD40L.

Reviewer #2:

Remarks to the Author:

... This study provides an important insight into the airway pre-existing resident T cell populations in human subjects, but the extent to which these T cells are protective or what their role is in prevention of infection remains unclear.

Thank you; for comments on the lack of protection data, please see first paragraph of response to Reviewer 1.

Specific comments:

1. In the conclusion section of the article, the authors write "The data presented here provide an explanation for the observed association between polymerase-cross-reactive T cells and rapidly aborted infection, representing an innate-like immediate response at the site of infection and potentially amplifying subsequent memory responses".

While the existence of cross-reactive TRM in the airways is a potential explanation for abortive infections, no evidence was shown in this study regarding the innate-like function of these T cells, as well as amplification of memory responses following reactivation. Besides the production of IFN γ and TNF α , could the authors comment on how does the activation of these T cells result in protection against virus?

The rapid CD4 T cell cytokine response that we demonstrate has been shown to mediate 'innate-like' induction of an antiviral state as well as recruitment of dendritic cells for CD8 priming in murine studies (*Zhao et al, Immunity 2016*), but we have removed this term and clarified that discussion of their potential antiviral functions is speculative and based on animal studies (Lines 188 and 194). Other studies have shown cytotoxic potential for CD4s in SARS-CoV-2 (*Meckiff et al, Cell 2020*) and they could also be supporting humoral immunity, although our previous study suggested a role for T cells in aborting infection in the absence of antibodies (*Swadling et al, Nature 2022*); these points have been added to the discussion (Line 197).

2. Related to point 1), it is of interest that the frequency of antigen-specific CD4 T cells in the lung is considerably greater than that of antigen-specific CD8 T cells (Figure 1). In fact, the total SARS-CoV-2 specific CD4 T cell response seems remarkably high (10% in Fig 1c)! How do the authors believe that these CD4 T cells contribute to control of virus? Is it through T cell help and rapid induction of nAbs? Or is it through some other mechanism?

Please see response above and response to Reviewer 1, point 2.

3. A key premise of the paper is that the SARS-CoV-2 specific T cells in the unexposed individuals are in fact cross-reactive T cells from prior exposure to other coronaviruses, presumably the common cold viruses. Could the authors provide some evidence for this? If mononuclear cells from BAL are stimulated with overlapping peptides from the spike proteins or RTC from the major common cold viruses, what would the frequencies of antigen-specific CD4 and CD8 T cells be?

We thank the reviewer for this excellent suggestion which has generated a compelling new dimension to our study. We have provided new data indicating that prior exposure to closely-related human seasonal coronaviruses is one likely source for the airway T cell reactivity observed, with stronger peripheral T cell and antibody responses to spike HCoV found in those with detectable airway T cells cross-reacting with SARS-CoV-2 (new Fig.4). It would be interesting to also check for T cell responses to HCoV in pre-pandemic BAL as suggested, but unfortunately there were insufficient cryopreserved vials left for this.

4. Lung TRM have been shown in several mouse infection models to wane over time (for e.g. in [1]) and several

other studies highlight the impermanence of the lung TRM compartment [2]. Given the potential waning of this cell population in the airways, what do the authors think in terms of the potential of lung TRM to induce long-lasting protection against respiratory infections?

This is an important point that we have added to the discussion, along with the suggested references. It is encouraging that a study of human lung T_{RM} showed stable persistence for more than a year in some transplant recipients (Snyder *et al*, *Sci. Immunol.* 2019), but more work needs to be done to examine whether interstitial lung T_{RM} perhaps need to specifically be induced to repopulate the airways and maintain longevity (Takamura *et al*, *JEM* 2019) or whether repeated mucosal vaccine boosters may be required (Line 213).

5. How representative is this 10-subject cohort of the general population? Could their previous challenge with live pneumococci (or the LAIV) have induced an environment in the lung that recruits more T cells in a non-specific manner simply due to the inflammation present in the lungs? Previous studies have shown lung TRM recruitment without the need for specific antigen when there is inflammation [3].

Our cohort is too small to extrapolate to the general population but interestingly, 6/10 of them having detectable pre-existing SARS-CoV-2 T cell responses is roughly consistent with household exposure and human challenge studies where 50% of the cohorts resisted infection (Kundu *et al*, *Nat Comms* 2022; Killingley *et al*, *Nat Med* 2022). We have added the important suggestion and reference regarding the possibility that prior inflammatory exposures such as pneumococcal challenge increased the recruitment of SARS-CoV-2-reactive lung T_{RM} specificities (Line 177). However these represent 'real-world' physiological exposures, with >10% of the general population being pneumococcal carriers (Almeida *et al*, *Journal of Infectious Diseases* 2021) and many receiving annual influenza vaccines. We have added a paragraph and new extended data addressing the caveat of the prior challenge of this cohort (Line 166), which was mostly more than 4 months before BAL sampling, see response to Reviewer 1, point 2.

[1] Slutter, B. *et al*. Dynamics of influenza-induced lung-resident memory T cells underlie waning heterosubtypic immunity. *Sci. Immunol.* 2, eaag2031 (2017)

[2] Zheng, M.Z.M., Wakim, L.M. Tissue resident memory T cells in the respiratory tract. *Mucosal Immunol* 15, 379–388 (2022). <https://doi-org.stanford.idm.oclc.org/10.1038/s41385-021-00461-z>

[3] Chung H, Kim EA, Chang J. A "Prime and Deploy" Strategy for Universal Influenza Vaccine Targeting Nucleoprotein Induces Lung-Resident Memory CD8 T cells. *Immune Netw.* 2021 Aug;21(4):e28. <https://doi.org/10.4110/in.2021.21.e28>

Reviewer #3:

Remarks to the Author:

A. This is an interesting and potentially important report that describes an assessment of T cell response to coronavirus peptides in lower airway T cells that were frozen prior to the SARS-CoV-2 pandemic. The team report large populations of cells that respond to these peptides and have features of tissue resident memory populations.

B. This is a novel and original report.

C. The quality of the presentation and writing is very strong. The experimental methodology is well described.

D. Appropriate. The work is not overstated and the limitation of the small sample size is clearly referenced.

E. The work is robust. The work relies on the specificity of the intracellular cytokine analysis after overnight peptide stimulation. The responses are clearly above DMSO background. It might be suggested that the use of a peptide pool from a heterologous antigen might have been useful in order to assess potential impact of non-specific activation but cell number is limited and the work is now complete. The increase from tissue cells compared to blood appears robust

F. I do not think that further experimental work is needed and assessment of these sample cohort is complete. Limitations in relation to interpretation reflect the small sample size.

We thank this reviewer for all these very positive comments and for recognising the limits of what was feasible with these precious samples from a small cohort.

The CD69 expression on peripheral blood lymphocytes is high at 7.6% of CD4+ cells and beyond what is normally considered as a recently activated population within peripheral blood. Do the authors have any comments on this ?

In our experience, the proportion of CD69 on peripheral T cells is analogous to what we find here if T cells are not pre-gated on memory markers CCR7 and CD62L in addition to CD45RO (Pallett *et al*, *JEM* 2017,

supplementary figure 1). In addition, we have shown that the proportion of CD69-expressing CD4 T cells depends on whether intermediate as well as CD69^{hi} populations are included in the gate (*Wiggins, Pallett, Li et al, Gut 2022*).

I am not sure if the title is quite right as it suggests the cells have been primed to target SARS-CoV-2 – “Airway-resident functional T cell responses against SARS-CoV-2 in unexposed individuals”. Perhaps “Airway-resident T cells with cross-reactive recognition of SARS-CoV-2 in unexposed individuals”.

We thank the reviewer for this excellent suggestion and have revised our title accordingly.

Minor typo in Figure 3 where (b) is labelled as (c).

Apologies for this error in the Fig.3 legend which has been corrected.

G. Appropriate

H. Very clear

All authors concur with the revised manuscript, which we hope you will now consider suitable for publication in Nature Immunology.

Decision Letter, first revision:**Subject:** Your manuscript, NI-LE33921A**Message:** Our ref: NI-LE33921A

14th Jul 2022

Dear Dr. Maini,

Thank you for your patience as we've prepared the guidelines for final submission of your Nature Immunology manuscript, "Airway-resident T cells from unexposed individuals cross-recognise SARS-CoV-2" (NI-LE33921A). Please carefully follow the step-by-step instructions provided in the attached file, and add a response in each row of the table to indicate the changes that you have made. Please also check and comment on any additional marked-up edits we have proposed within the text. Ensuring that each point is addressed will help to ensure that your revised manuscript can be swiftly handed over to our production team.

When you upload your final materials, please include a point-by-point response to any remaining reviewer comments and please make sure to upload your checklist.

If you have not done so already, please alert us to any related manuscripts from your group that are under consideration or in press at other journals, or are being written up for submission to other journals (see: <https://www.nature.com/nature-portfolio/editorial-policies/plagiarism#policy-on-duplicate-publication> for details).

In recognition of the time and expertise our reviewers provide to Nature Immunology's editorial process, we would like to formally acknowledge their contribution to the external peer review of your manuscript entitled "Airway-resident T cells from unexposed individuals cross-recognise SARS-CoV-2". For those reviewers who give their assent, we will be publishing their names alongside the published article.

Nature Immunology offers a Transparent Peer Review option for new original research manuscripts submitted after December 1st, 2019. As part of this initiative, we encourage our authors to support increased transparency into the peer review process by agreeing to have the reviewer comments, author rebuttal letters, and editorial decision letters published as a Supplementary item. When you submit your final files please clearly state in your cover letter whether or not you would like to participate in this initiative. Please note that failure to state your preference will result in delays in accepting your manuscript for publication.

Cover suggestions

As you prepare your final files we encourage you to consider whether you have any images or illustrations that may be appropriate for use on the cover of Nature

Immunology.

Nature Immunology has now transitioned to a unified Rights Collection system which will allow our Author Services team to quickly and easily collect the rights and permissions required to publish your work. Approximately 10 days after your paper is formally accepted, you will receive an email in providing you with a link to complete the grant of rights. If your paper is eligible for Open Access, our Author Services team will also be in touch regarding any additional information that may be required to arrange payment for your article.

Please note that *Nature Immunology* is a Transformative Journal (TJ). Authors may publish their research with us through the traditional subscription access route or make their paper immediately open access through payment of an article-processing charge (APC). Authors will not be required to make a final decision about access to their article until it has been accepted. [Find out more about Transformative Journals](https://www.springernature.com/gp/open-research/transformative-journals).

If you have any questions about costs, Open Access requirements, or our legal forms, please contact ASJournals@springernature.com.

Please use the following link for uploading these materials: [REDACTED]

Best regards,

Elle Morris
Senior Editorial Assistant
Nature Immunology
Phone: 212 726 9207
Fax: 212 696 9752
E-mail: immunology@us.nature.com

On behalf of

Jamie D.K. Wilson, D.Phil
Chief Editor
Nature Immunology
212 726 9207
j.wilson@us.nature.com

Reviewer #1:
Remarks to the Author:
I am satisfied with author's respond to my comments/concerns.

Reviewer #2:
Remarks to the Author:
The authors have addressed all my concerns.

Final Decision Letter:

Subject: Decision on Nature Immunology submission NI-LE33921B

Message: In reply please quote: NI-LE33921B

Dear Dr. Maini,

I am delighted to accept your manuscript entitled "Airway-resident T cells from unexposed individuals cross-recognise SARS-CoV-2" for publication in an upcoming issue of Nature Immunology.

Over the next few weeks, your paper will be copyedited to ensure that it conforms to Nature Immunology style. Once your paper is typeset, you will receive an email with a link to choose the appropriate publishing options for your paper and our Author Services team will be in touch regarding any additional information that may be required.

Please note that *Nature Immunology* is a Transformative Journal (TJ). Authors may publish their research with us through the traditional subscription access route or make their paper immediately open access through payment of an article-processing charge (APC). Authors will not be required to make a final decision about access to their article until it has been accepted. [Find out more about Transformative Journals](https://www.springernature.com/gp/open-research/transformative-journals).

Your paper will be published online soon after we receive your corrections and will appear in print in the next available issue. Content is published online weekly on Mondays and Thursdays, and the embargo is set at 16:00 London time (GMT)/11:00 am US Eastern time (EST) on the day of publication. Now is the time to inform your Public Relations or Press Office about your paper, as they might be interested in promoting its publication. This will allow them time to prepare an accurate and satisfactory press release. Include your manuscript tracking number (NI-LE33921B) and the name of the journal, which they

will need when they contact our office.

About one week before your paper is published online, we shall be distributing a press release to news organizations worldwide, which may very well include details of your work. We are happy for your institution or funding agency to prepare its own press release, but it must mention the embargo date and Nature Immunology. Our Press Office will contact you closer to the time of publication, but if you or your Press Office have any enquiries in the meantime, please contact press@nature.com.

Also, if you have any spectacular or outstanding figures or graphics associated with your manuscript - though not necessarily included with your submission - we'd be delighted to consider them as candidates for our cover. Simply send an electronic version (accompanied by a hard copy) to us with a possible cover caption enclosed.

Please note that we encourage the authors to self-archive their manuscript (the accepted version before copy editing) in their institutional repository, and in their funders' archives, six months after publication. Nature Portfolio recognizes the efforts of funding bodies to increase access of the research they fund, and strongly encourages authors to participate in such efforts. For information about our editorial policy, including license agreement and author copyright, please visit www.nature.com/ni/about/ed_policies/index.html

Sincerely,

Jamie D.K. Wilson, D.Phil
Chief Editor
Nature Immunology
212 726 9207
j.wilson@us.nature.com